ⓐ | **Open Peer Review** | Bacteriology | Research Article

# Characterization of YdgH: a mediator of beta-lactam susceptibility in Enterobacterales

Jacob E. Lazarus,[1] Matthew K. Waldor,[2,3,4] David C. Hooper[1]

**ABSTRACT** Beta-lactam antibiotics are often the treatment of choice for serious bacterial infections. In a previous screen for novel genetic mediators affecting beta-lactam susceptibility, we discovered that deletion of *ydgH*, a conserved gene of unknown function, leads to increased resistance to beta-lactams, as well as increased susceptibility to detergent compounds. Here, we further characterize YdgH in *Serratia marcescens*, *Enterobacter cloacae*, and *Escherichia coli* using a combination of biochemical and cell biological approaches. We find that YdgH fractionates with periplasmic proteins, and this periplasmic localization is necessary for its function. Using purified recombinant protein, we demonstrate that YdgH is a relatively compact, globular monomer. The YdgH polypeptide contains three tandem DUF1471 domains. In a Δ*ydgH* background, overexpression of polypeptides containing both the second and the third, but not the first DUF1471 domain, is necessary to rescue the deletion phenotype. To determine how YdgH function influences beta-lactam and detergent susceptibility, we tested several targeted hypotheses. We found that deletion of *ydgH* neither affects *ompC* or *ompF* transcript levels, nor does it alter the processing of lipopolysaccharide, nor does it activate the sigma E regulon alone or in combination with mutations in other periplasmic proteins. Finally, we delineate the results of a genetic screen for spontaneous mutants that complement the detergent susceptibility phenotype, the results of which may fuel the further studies that are necessary to determine the precise role YdgH plays in bacterial physiology.

**IMPORTANCE** Beta-lactams such as penicillins and cephalosporins are the most commonly prescribed antibiotics for serious bacterial infections. Increasing antibiotic resistance threatens their effectiveness. We previously identified the uncharacterized gene *ydgH* as a modifier of beta-lactam susceptibility in Gram-negative bacteria. To begin to understand the specific role of YdgH, in this study, we perform initial characterizations of this protein. We also test hypotheses as to how the function of YdgH contributes to beta-lactam physiology.

**KEYWORDS** beta-lactam, cell envelope, periplasm, outer membrane

**Peer Reviewer** Arun S. Kharat, Jawaharlal Nehru University, New Delhi, India

Address correspondence to Jacob E. Lazarus, Jacob.Lazarus@mgh.harvard.edu.

The authors declare no conflict of interest.

See the funding table on p. 13.

The Gram-negative bacterial cell envelope, consisting of the inner membrane, the periplasmic space and its associated peptidoglycan, and the outer membrane and its associated lipopolysaccharide (LPS), is a critical structure that maintains cell integrity and allows regulated interaction with the environment (1). It is the target of several major classes of antibiotics, most notably beta-lactams, which act on peptidoglycan and account for nearly two-thirds of worldwide antibiotic use (2). The emergence of resistance to beta-lactams underscores the importance of understanding the genetic and molecular mechanisms that govern cell envelope integrity and antibiotic susceptibility in Gram-negative pathogens (3, 4). In a previous transposon insertion sequencing mutagenesis screen for novel mediators of beta-lactam susceptibility in *Serratia*

*marcescens*, we identified *ydgH*, a gene of unknown function (5). Characterization of antimicrobial phenotypes revealed that *ydgH* deletion mutants had decreased susceptibility to several beta-lactam antibiotics and increased susceptibility to both cationic and anionic detergents, suggesting that YdgH might play a role in cell envelope maintenance.

YdgH is a member of a small group of structurally related, domain of unknown function 1471 (DUF1471)-containing proteins that are present in the Enterobacterales. DUF1471-containing proteins can be divided into 12 paralogous families, with any one species of bacteria containing around 5–10 family members. These family members, typically made up of single DUF1471 domains, have been implicated in diverse cellular processes, most broadly characterizable as involved in stress responses, biofilm formation, or pathogenicity (6), although they have not been shown to interact genetically or biochemically with each other, or to be involved in a single discrete pathway.

Data on the function of YdgH specifically are sparse. YdgH contains a predicted periplasmic signal sequence, like other DUF1471-family members, but is unique in containing three tandem DUF1471 domains, rather than a single domain. The *Salmonella enterica* homolog was identified as a putative secreted virulence factor based on proteomics on culture supernatants (7), but it was only detected in low abundance in samples from mutants deleted for a key type III secretion system regulatory protein and not in wild-type samples. Efforts to identify cognate eukaryotic targets were unsuccessful (8). High-confidence prokaryotic binding partners have not been identified, but high-throughput screens have identified a small number of unconfirmed putative interactors (9, 10).

Here, we report further characterization of YdgH and targeted experimentation to investigate its previously observed beta-lactam and detergent phenotypes.

## MATERIALS AND METHODS

### Bioinformatics and structural modeling

YdgH homologs were identified using a standard protein BLAST using *S. marcescens* YdgH (WP_033640181). Homologs were not present in non-Enterobacterales. The YdgH amino acid alignment was constructed through the phylogenetic analysis feature of MEGA X (11). The GenBank entries used were as follows: *S. marcescens* WP_033640181; *Serratia liquefaciens* WP_129940662; *Yersinia enterocolitica* WP_083158798; *Hafnia alvei* WP_072310388; *Pantoea agglomerans* WP_109650442; *Escherichia coli* K12 WP_000769303; *Shigella dysenteriae* WP_128880534; *E. coli* O157:H7 H&_AAG586591; *Citrobacter freundii* WP_038640470; *S. enterica* WP_079974581; *Enterobacter cloacae* WP_088221952; *Raoultella planticola* WP_064793623; *Klebsiella aerogenes* WP_047035392; *Klebsiella pneumoniae* WP_110096565; *Chronobacter sakazakii* WP_110868768. The periplasmic signal sequence was predicted using SignalP 6.0 (12) and the boundaries of the DUF1471 domains using AlphaFold and Robetta (13, 14). The two putative YdgH conformations were predicted by Robetta and prepared for depiction with Jena3d (15).

### Molecular biology

Cloning was performed using primers from either Integrated DNA Technologies (Coralville, IA, USA) or Eton Bio (Boston, MA, USA), genomic DNA or Gblocks from Integrated DNA Technology, Q5 polymerase, and HiFi DNA Assembly (New England Biolabs, Ipswich, MA, USA). Transformations were performed via electroporation. The full amino acid sequence of the C-terminal Strep-Strep-FLAG tag was GGWSHPQFEKGGGSGGGSGGGSWSHPQFEKGASGEDYKDDDDK. Allelic exchange was performed with the pTOX system as previously described (16). The rescue plasmid was pBAD33. P1 lambda phage transduction was performed using standard methods and after (17).

## Recombinant protein experiments

Recombinant YdgH constructs were produced using the pET system in *E. coli* BL21. Constructs were expressed with an N-terminal 6xHis-TEV-SUMO tag. Lysis was with Emulsiflex (Avestin, Ottawa, Canada). After clarification of the soluble fraction by centrifugation, recombinant protein was captured on nickel resin and then eluted with SUMO protease (Thermo Fisher Scientific, Waltham, MA, USA). Protein was subsequently exchanged into size exclusion buffer (20 mM Tris pH 7.8, 150 mM NaCl, 10% glycerol) using a PD10 column (Thermo Fisher Scientific). Small aliquots were flash frozen in liquid nitrogen. The traces depicted are the average of three size exclusion runs. Bio-Rad (Hercules, CA, USA) gel filtration standards (thyroglobulin, gamma-globulin, ovalbumin, myoglobulin, and vitamin B12) were used for the calibration of the Superdex 200 10/300 Increase column on which samples were run. Glutaraldehyde was used at a final concentration of 0.5%, and the reaction was stopped after 10 minutes with excess glycine.

## Bacteriology

In all liquid growth assays, bacteria were grown at 37°C with orbital shaking at 275 rpm in lysogeny broth (LB). pBAD33 was induced with 1% arabinose upon dilution from overnight culture. Osmotic shock and subcellular fractionation were performed after (18), with the final ultracentrifugation step omitted. Cefoxitin was diluted to 1.25 µg/mL for experiments with *E. coli* BW25113 and 8 µg/mL for experiments with *S. marcescens*. For sigma E induction experiments, a simplified Miller method was used after (19), with ONPG (Gold Biotechnology, St. Louis, MO, USA) as substrate. For the selection of *ydgH* detergent revertants, benzethonium chloride was used in LB agar at a concentration of 125 µg/mL. Revertants were observed at a frequency of around 1 per 150,000 plated CFU. After verifying wild-type benzethonium chloride MICs, colonies were submitted to SeqCenter (Pittsburgh, PA, USA) for genomic DNA extraction, whole genome sequencing, and variant calling (using breseq [20]). LPS was isolated from *E. cloacae* using LPS Extraction Kit (Bulldog Bio, Portsmouth, NH, USA) and, after electrophoresis, stained using ProQ Emerald 300 Lipopolysaccharide Gel Stain Kit (Thermo Fisher Scientific) and visualized using a Bio-Rad Gel Doc EZ.

## Relative quantitative RT-PCR assays

To determine relative *ompF* and *ompC* transcript levels, total *E. coli* and *S. marcescens* RNA was isolated from log-phase cultures using the RNeasy Midi Kit (Qiagen) according to the manufacturer's directions, followed by reverse transcription using the Verso cDNA synthesis kit (Abgene, Thermo Fisher). Real-time reverse transcription relative quantitative PCR was performed using EvaGreen dye on a CFX Opus 96 Real-Time PCR System (Bio-Rad). *mdh* was used as a housekeeping gene control.

## Materials

Cefoxitin was obtained from Sigma-Aldrich (St. Louis, MO, USA). CAG45115 was generously provided by the laboratory of Carol Gross (University of California, San Francisco), and P1 lambda phage by the laboratory of Thomas Bernhardt (Harvard University). Anti-FLAG, -MalE, -RpoB, and Omp antibodies were obtained from Abcam (Waltham, MA, USA) and, after labeling with a horseradish peroxidase (HRP)-conjugated secondary antibody, visualized with a digital chemiluminescence LCD imager. Prior to loading gels, total protein concentration was quantitated with Pierce BCA Assay (Thermo Fisher Scientific) or by the integration of the total Coomassie stainable protein, and an approximately equal amount was loaded in each lane.

## RESULTS

### Subcellular localization of YdgH

We began by aligning the *S. marcescens ydgH* gene against its Enterobacterial homologs (Fig. 1). A high-confidence (98%) periplasmic signal sequence was identified by SignalP 6.0 at residues 1–22 (12). To validate this prediction and to determine if this predicted periplasmic signal sequence mediated periplasmic localization, we introduced a C-terminal Strep-Strep-FLAG (SS-FLAG)-tagged gene into the *S. marcescens* ATCC 13880 native genomic locus using allelic exchange. We have previously established that the Δ*ydgH* mutant has increased resistance to cefoxitin, as measured by increased growth in liquid media (operationalized as the ratio of mutant to wild-type $OD_{600}$ at 4 hours). When introduced into the deletion background, the SS-FLAG-tagged gene restored the wild-type cefoxitin phenotype (Fig. 2A), indicating that the tagged YdgH protein was functional. Using this strain, we performed subcellular fractionation after osmotic shock and spheroplasting, confirming that YdgH localizes to the periplasm (Fig. 2B). To determine if this localization was necessary for YdgH function, we transformed Δ*ydgH* with a plasmid encoding either full-length FLAG-tagged *ydgH* or FLAG-tagged *ydgH* without its secretion signal (Fig. 2C). While overexpression of wild-type YdgH in the YdgH deletion background rescued the cefoxitin phenotype, overexpression of a YdgH construct lacking the secretion signal did not (Fig. 2D), suggesting that the periplasmic localization of YdgH is necessary for its function.

### Biochemical characterization of YdgH

While individual, isolated YdgH domains have been crystallized and are known to adopt the characteristic, compact DUF1471 fold (6), the overall arrangement of the three DUF1471 domains (which we designate DUF1471-A, DUF1471-B, and DUF1471-C, Fig. 1) with respect to each other in YdgH is unknown. To begin, we modeled the overall polypeptide in AlphaFold, which predicted that there was a relatively large region without predicted structure between DUF1471-B and DUF1471-C (13). As an initial hypothesis of the possible conformations the YdgH DUF1471 domains might make around this unstructured region, we explored possible structures in Robetta (14). This analysis revealed a possible conformation with DUF1471-B in close proximity to DUF1471-C, generating a compact surface (Fig. 3A) and another where DUF1471-C hinges about the unstructured region, generating a more elongated surface (Fig. 3B).

To begin to differentiate between these possibilities, we purified recombinant full-length YdgH (without a signal sequence, "22-Cterm") and polypeptides (Fig. 4A) with C-terminal truncations encoding: DUF1471-A, -B, and the unstructured region ("22–244"), DUF1471-A and -B ("22–188"), and DUF1471-A ("22–100"). We performed size exclusion chromatography on these polypeptides, and by comparing their elution to globular size standards, computed $R_S/R_{min}$ (the calculated Stokes radius compared to the minimal Stokes radius of a perfectly spherical protein) (Fig. 4B). We found that all polypeptides had an $R_S/R_{min}$ less than 1.3, suggesting an overall globular fold of YdgH, perhaps similar to the compact surface in Figure 3A. This also suggested that YdgH is monomeric, similar to the oligomerization state previously observed for purified isolated DUF1471 domains (6). To confirm its monomeric nature, we subjected full-length YdgH to glutaraldehyde cross-linking, using bovine serum albumin (BSA) and IgG as monomeric and oligomeric controls, respectively. We observed minimal oligomerization (Fig. 4C), indicating that full-length YdgH is also monomeric.

### Functional analysis of DUF1471 domains

We next set out to determine which YdgH domain(s) is required for its native function. We generated a small panel of FLAG-tagged expression plasmids, each encoding a YdgH truncation construct (and an N-terminal secretion signal) (Fig. 5A). After transforming a wild-type expression construct or one of the truncation constructs into Δ*ydgH*, we then assayed for either wild-type or mutant susceptibility to cefoxitin. The YdgH constructs

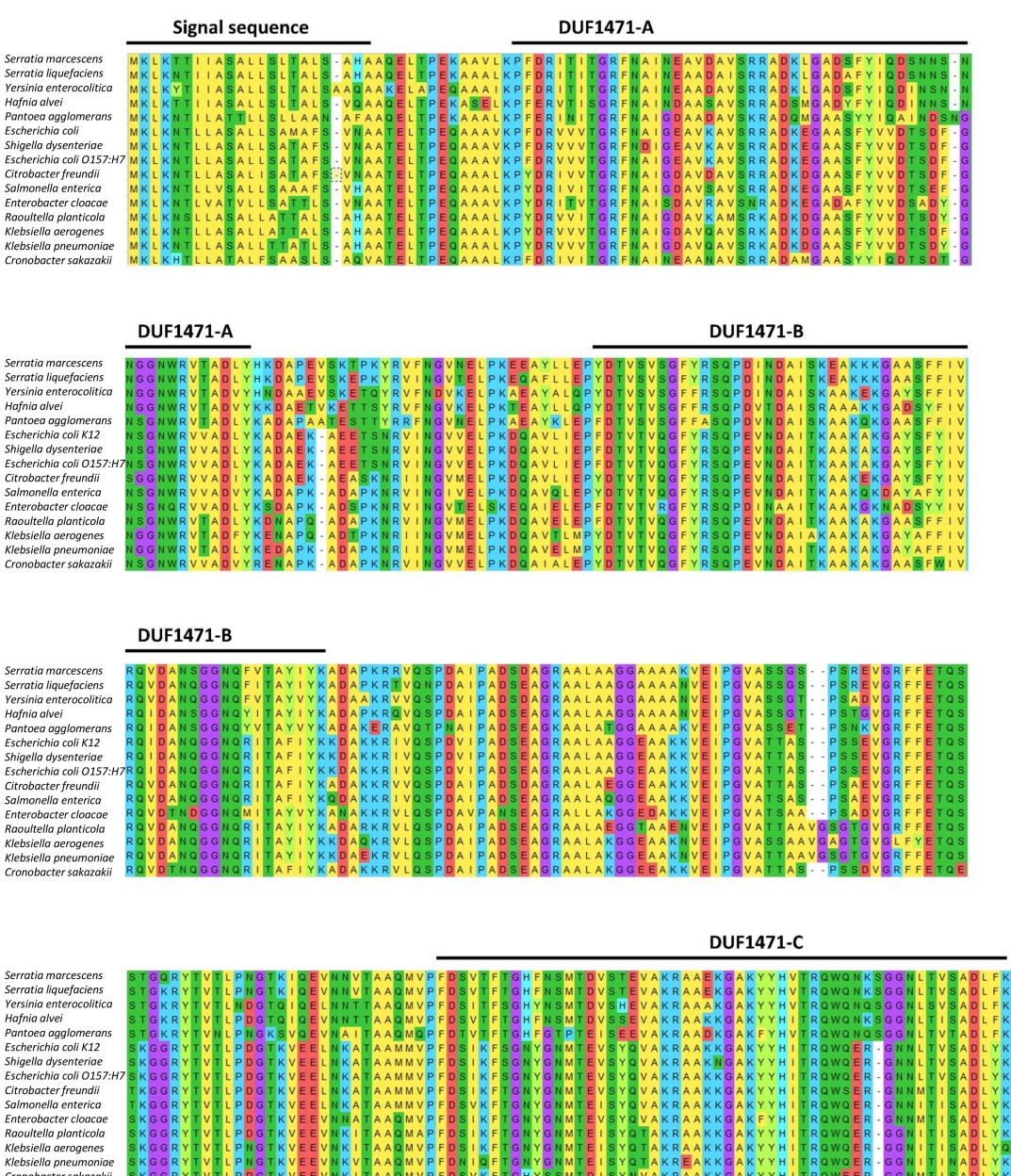

**FIG 1** Phylogenetic analysis based on translated amino acids was performed in MEGA X. Periplasmic signal sequences were predicted in SignalP and DUF1471 domains in AlphaFold and Robetta. Genbank entries: SM WP_033640181; SL WP_129940662; YE WP_083158798; HA WP_072310388; PA WP_109650442; ECK12 WP_000769303; SD WP_128880534; ECO157 H&_AAG586591; CF WP_038640470; SE WP_079974581; ECLO WP_088221952; RP WP_064793623; KA WP_047035392; KP WP_110096565; CS WP_110868768.

had variable levels of expression, with a construct containing only an isolated DUF1471 domain expressing poorly (Fig. 5B). Expression was also minimal with a construct containing DUF1471-A and -B without the unstructured region between DUF1471-B and -C. Despite good levels of expression, constructs lacking either DUF1471-B ("187C") or DUF1471-C ("N244") were not able to restore the wild-type cefoxitin phenotype (Fig. 5C). In contrast, expression of a construct encoding DUF1471-B, -C, and the unstructured region between them ("101C") was sufficient, indicating that DUF1471-A is dispensable for this function of YdgH.

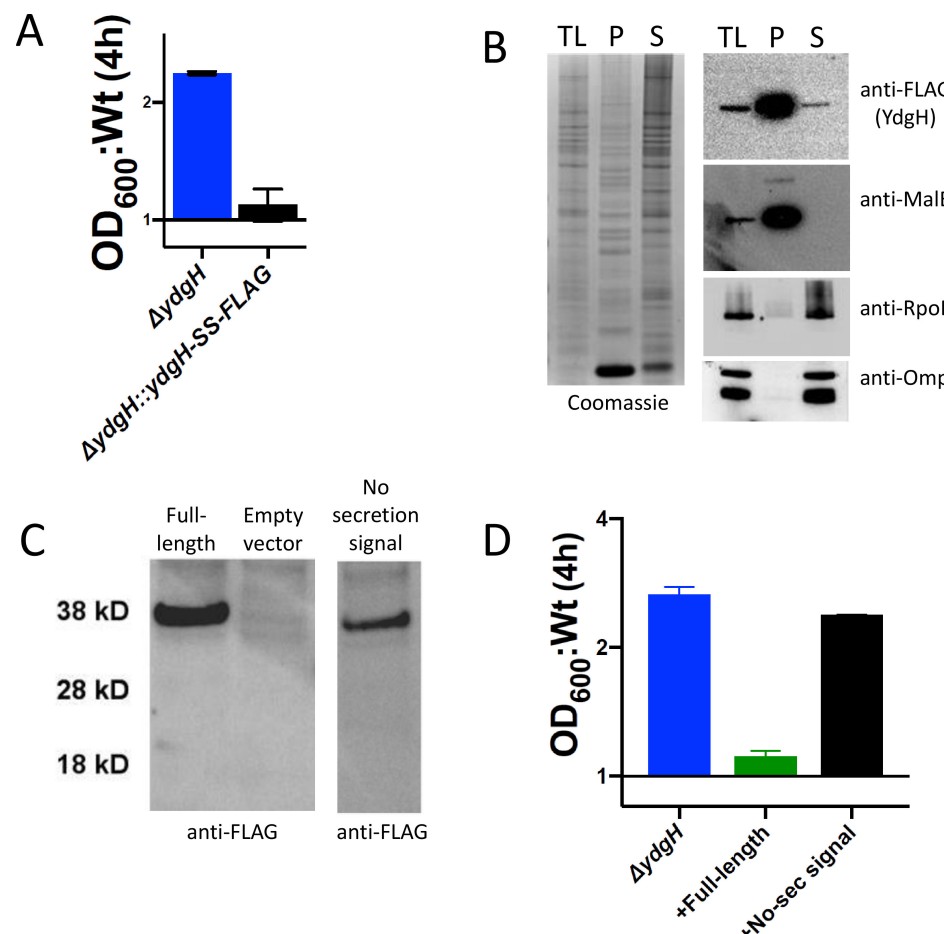

**FIG 2** Subcellular localization of YdgH. (A) In a Δ*ydgH* background, insertion of a Strep-Strep-FLAG-tagged coding sequence at the genomic locus restores wild-type cefoxitin phenotype. (B) Subcellular fractionation after osmotic shock reveals YdgH localizes to the periplasm. Left panel, Coomassie-stained total lysate, periplasm, and spheroplast fractions. Right panel, Western blots for YdgH (anti-FLAG), MalE (a periplasmic protein), RpoB (a cytoplasmic protein), and outer membrane proteins (membrane-associated proteins). TL, total lysate; P, periplasm; S, spheroplast fraction. (C) Western blot of FLAG-tagged rescue constructs expressed in a Δ*ydgH* background. (D) In a Δ*ydgH* background, rescue with a plasmid-expressed full-length FLAG-tagged YdgH but not one without a secretion signal restores wild-type cefoxitin phenotype.

## Investigations of the role of YdgH in the cell envelope

We began our investigation into the mechanism by which YdgH functions with the hypothesis that YdgH acts in the periplasm in a pathway that likely facilitates outer membrane integrity. Several observations informed this hypothesis. First, disruption of YdgH leads to detergent susceptibility, and detergents are thought to act primarily at the bacterial outer membrane. Second, an examination of the region 5′ to the *ydgH* start codon revealed a potential sigma 24 (sigma E) promoter that has previously been verified in a high-throughput screen (21, 22); sigma E activates transcription of its target genes in response to a diverse array of envelope stresses (1). Finally, beta-lactam antibiotics like cefoxitin must permeate through the outer membrane to the periplasm to reach their target, so disruption of this process could lead to a decrease in periplasmic beta-lactam concentration.

The major route of permeation for beta-lactams is through outer membrane porins, primarily OmpF and OmpC (23). We examined *ompC* and *ompF* transcript levels in *S. marcescens* and in *E. coli* BW25113 (the parent strain of the KEIO gene knockout collection) (24). We first confirmed that the *E. coli* BW25113 Δ*ydgH* mutant recapitulated the phenotype from *S. marcescens* (Fig. 6A). We then compared expression levels of

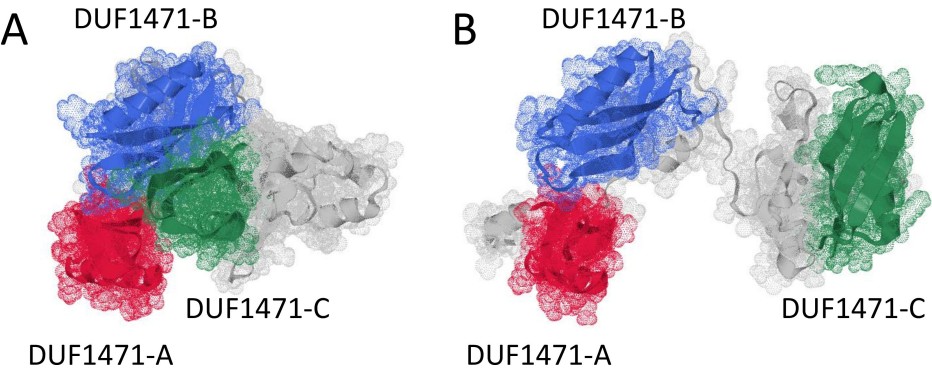

**FIG 3** Structural predictions of YdgH tertiary structure. (A) A compact predicted structure with the three DUF1471 domains in close approximation to each other and (B) a more elongated predicted structure with DUF1471-C hinged about the unstructured region C-terminal to DUF1471-B.

*ompC* and *ompF* transcripts between BW25113 WT and Δ*ydgH* mutant, as well as in *S. marcescens*; in neither species were *ompC* or *ompF* levels affected by *ydgH* disruption (Fig. 6B), suggesting that the beta-lactam phenotype is likely not due to quantitative differences in outer membrane porins. Like Δ*ydgH*, mutants defective in LPS elaboration are also known to have increased susceptibility to detergents and to have decreased susceptibility to cephalosporins (25). We used the previously characterized *E. cloacae* ATCC 13047 Δ*ydgH* mutant (5) for these experiments, as *E. coli* BW25113 and *S. marcescens* ATCC 13880 did not have well-defined O antigen laddering. However, we did not detect either quantitative differences in total stainable LPS or qualitative differences in the O antigen laddering pattern in *E. cloacae* (Fig. 7).

As a more general readout of periplasmic and outer membrane stress, we assayed whether in addition to being a potential sigma E target, disruption of *ydgH* leads to the activation of the sigma E regulon itself, either alone or in combination with disruption of other periplasmic and membrane homeostasis pathways. We utilized a previously described *E. coli* strain, CAG45114, with a beta-galactosidase reporter for sigma E activation (26). We began by generating a strain with *ydgH* disrupted by an ampicillin resistance cassette. This allowed us, using phage transduction, to insert this cassette into either the wild-type CAG45114 strain or a group of CAG45114 strains transduced with kanamycin resistance cassettes from the KEIO collection targeting core periplasmic and membrane homeostasis pathways. We found that neither CAG45114 *ydgH::ampR* nor *ydgH::kan* displayed sigma E activation when compared to wild type or *rseB::kan* control (Fig. 8). *ydgH* double mutants did not display additive activation in sigma E in pathways involved in disulfide bond formation and isomerization (*dsbA, dsbC*), protein folding and quality control (*fkpA, degP*), outer membrane protein assembly and insertion (*bamE*), or LPS processing (*rfaB, rfaG*), suggesting that *ydgH* may not participate in these pathways.

As a less targeted approach to discover pathways in which YdgH might participate, we constructed an expanded panel of single and double mutants in BW25113 and analyzed their growth in LB broth to assay for genetic interaction. We did not observe compound fitness defects when compared to single mutants in the following genes: *ompR, ompF, ompC, ompN, phoE, phoP, degP, fkpA, ppiD, ppiA, dsbC, dsbA, lpxM, rfaG, rfaQ, rfaP, galU, metQ, fepE, amiA, rcsF, marA, fur, rseB, bamE, nlpl, kdsC, pgpA, ygbE, ybiS, ytfJ, ytfK, yjfN, ybfA, yhcN,* or *bsmA* (data not shown). To help formulate additional hypotheses, we utilized the increased susceptibility of *E. coli* BW25113 Δ*ydgH* to detergent, and generated escape mutants (spontaneous mutants with restored wild-type susceptibility) on benzethonium chloride agar. We performed whole genome sequencing and in independent mutants, identified single mutations in the coding regions of *proQ, ftsI, mepM, fadR, yhbJ, pnp, ftsA, lpxM,* as well as in the *psrO* sRNA. We identified two independent mutants with different mutations in the coding region of *yhdP*. We also identified a mutation in the intergenic region between *ykgM* and *ykgR*. In one escape

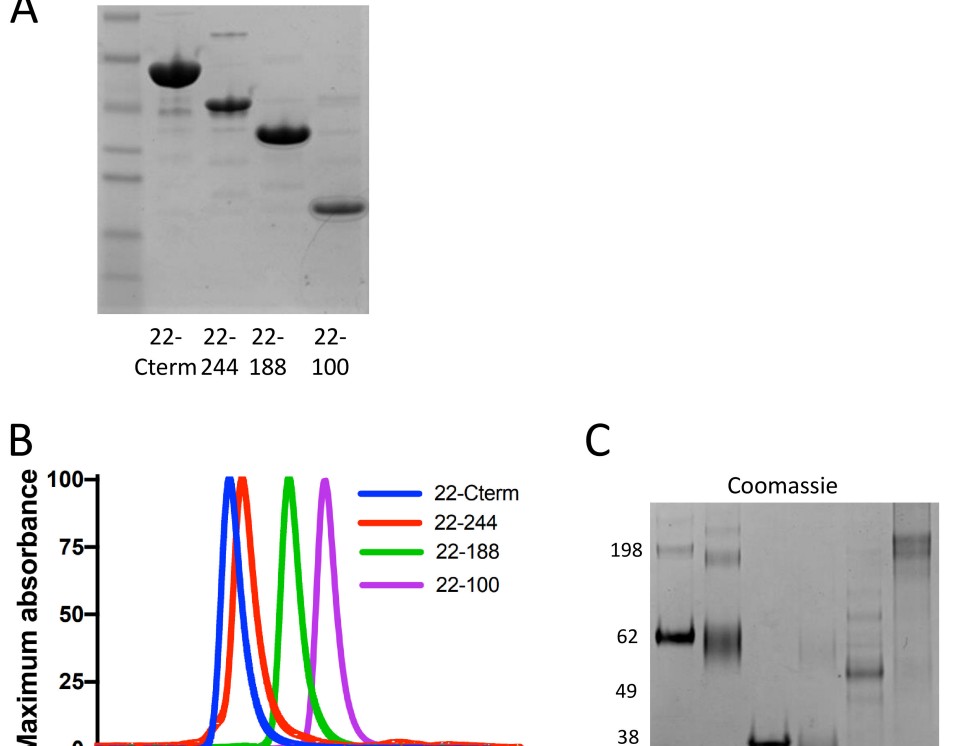

**FIG 4** YdgH may be a compact monomer. (A) Purified recombinant YdgH polypeptides (sans secretion signal). (B) Size exclusion chromatographs on Superdex 200 10/300 Increase column of indicated recombinant polypeptides (top) and calculated hydrodynamic parameters (bottom). (C) Glutaraldehyde cross-linking of (predominantly) monomeric bovine serum albumin (BSA), YdgH (22-Cterm), and the oligomeric IgG.

mutant, mutations in the coding regions of *arnB, envZ,* and *malS* were identified. The specific mutations observed are detailed in Table 1.

## DISCUSSION

In this study, we have performed the initial characterization of YdgH, which we have previously identified as a novel modifier of beta-lactam and detergent susceptibility. Here, using subcellular fractionation and rescue experiments, we have confirmed that YdgH localizes to the periplasm, and this localization is necessary for its function. In size exclusion chromatography and cross-linking experiments, we show that YdgH is monomeric, and our early analysis here leads us to speculate that it may form a compact, rather than a more elongated tertiary structure. Of the three DUF1471 domains, the second and the third are necessary and sufficient for wild-type beta-lactam susceptibility.

It seems tempting to speculate that YdgH, in the periplasm, participates in a pathway that contributes to envelope stability, because that is where the targets of both beta-lactam and detergent are, especially considering the sigma E promoter site upstream of *ydgH*. We did not detect quantitative differences in Δ*ydgH* either for the levels of *ompC* and *ompF* transcripts, or in the electrophoretic banding patterns of LPS. Using sigma E activation as a readout, we performed a targeted screen for genetic interactors of *ydgH*. Disruption of *ydgH* itself did not lead to sigma E activation. We hypothesized

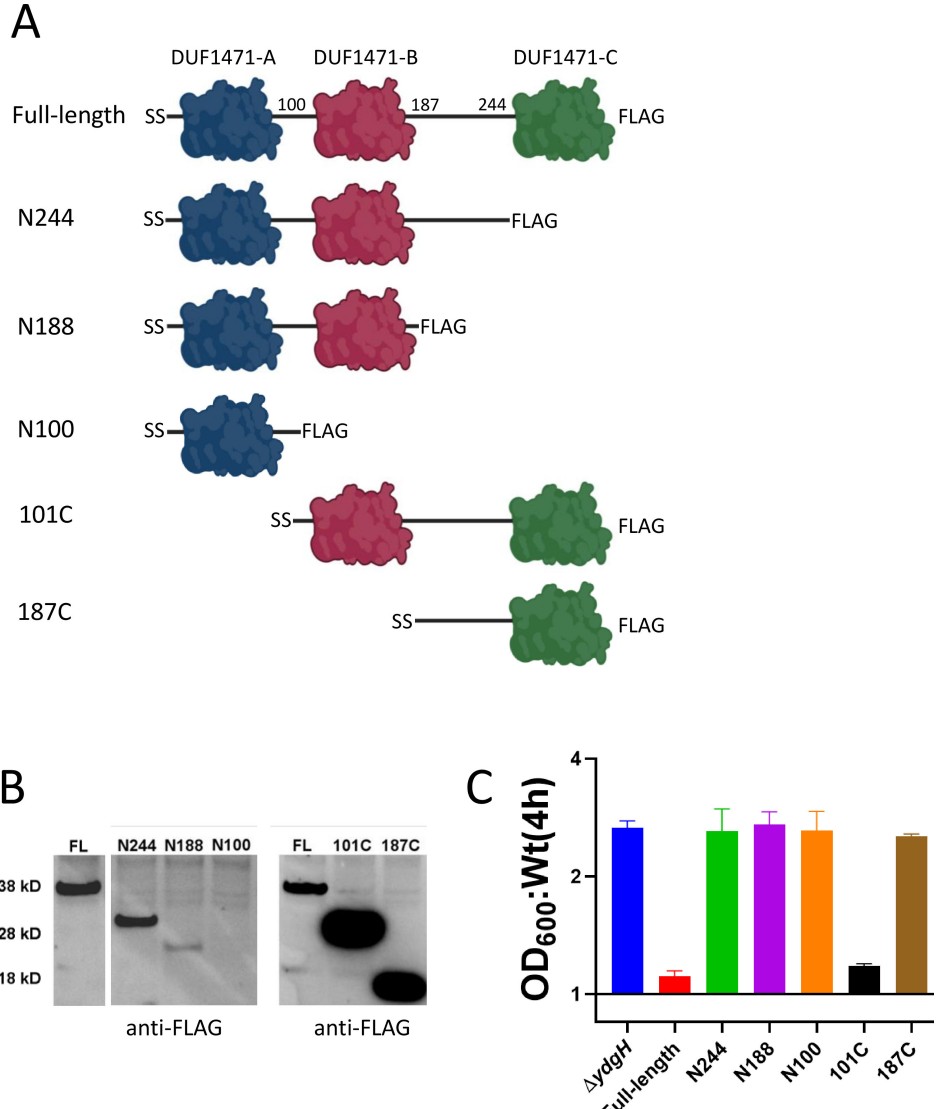

**FIG 5** DUF1471-B and DUF1471-C are required for YdgH function. (A) C-terminal FLAG-tagged YdgH constructs used in rescue experiments. (B) Expression of rescue constructs. Strains carrying rescue constructs were induced with the same concentration of inducer. (C) Expression of full-length YdgH and 101C (the construct encoding DUF1471-B, DUF1471-C, and the unstructured region between them) restores wild-type cefoxitin phenotype.

that redundancies in *ydgH* function might be revealed in double mutants. There was no additive activation of sigma E in double mutants with disruptions in periplasmic protein folding, outer membrane beta-barrel insertion, or in LPS biosynthesis, suggesting, along with our other experiments, that YdgH does not function in these pathways. We did not identify compound fitness defects in an expanded panel of double mutants, including those involved in osmoregulation and phosphate regulation, iron transport, peptidoglycan remodeling, capsule synthesis, or phospholipid homeostasis (27–32), nor in proteins identified as YdgH binding partners in high-throughput screens (FepE, YtfK, OmpN) (9, 10), nor in other periplasmic DUF1471-containing proteins (BsmA, YhcN, YjfN) (6). Additional experiments that could be performed to assess for mild defects in outer membrane integrity, either in single or in compound mutants, include fluorescent dye penetration and periplasmic enzyme leakage assays.

In which pathway might YdgH participate, then? High-confidence protein–protein interactions have not been identified in the literature, and in our initial attempts, we have

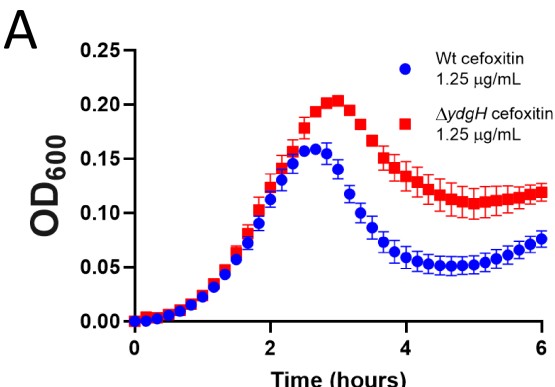

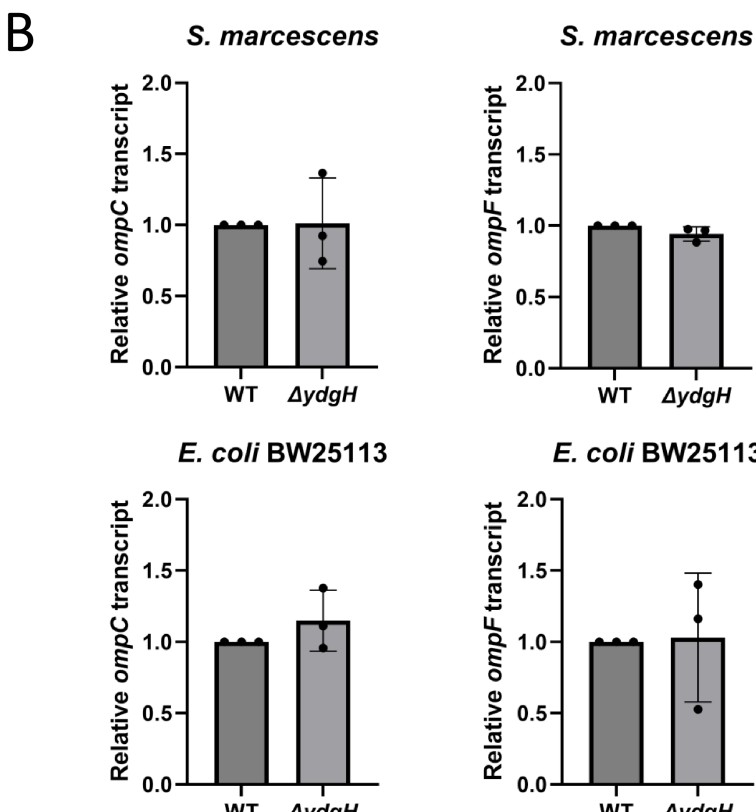

**FIG 6** Characterization of *omp* transcript levels. (A) *E. coli* BW25113 Δ*ydgH* recapitulates the cefoxitin phenotype seen in other Enterobacterales. (B) Real-time reverse transcription relative quantitative PCR of *ompC* and *ompF* in the indicated strains.

also not identified a compelling binding partner. It is possible that these interactions may be transient, and stabilizing them, perhaps with cross-linking, may be necessary. Interestingly, in our detergent revertant screen, three genes (*mepM, ftsA, ftsI*) involved in peptidoglycan homeostasis were found to have point mutations. Additionally, the periplasmic protease *prc*, known to be involved in peptidoglycan homeostasis, when deleted, has been found to have similar detergent and beta-lactam phenotypes to *ydgH* (25); in preliminary attempts, we have been unable to create a *prc* and *ydgH* double mutant. Future investigation of the peptidoglycan structure and muropeptide pool in Δ*ydgH* may be valuable.

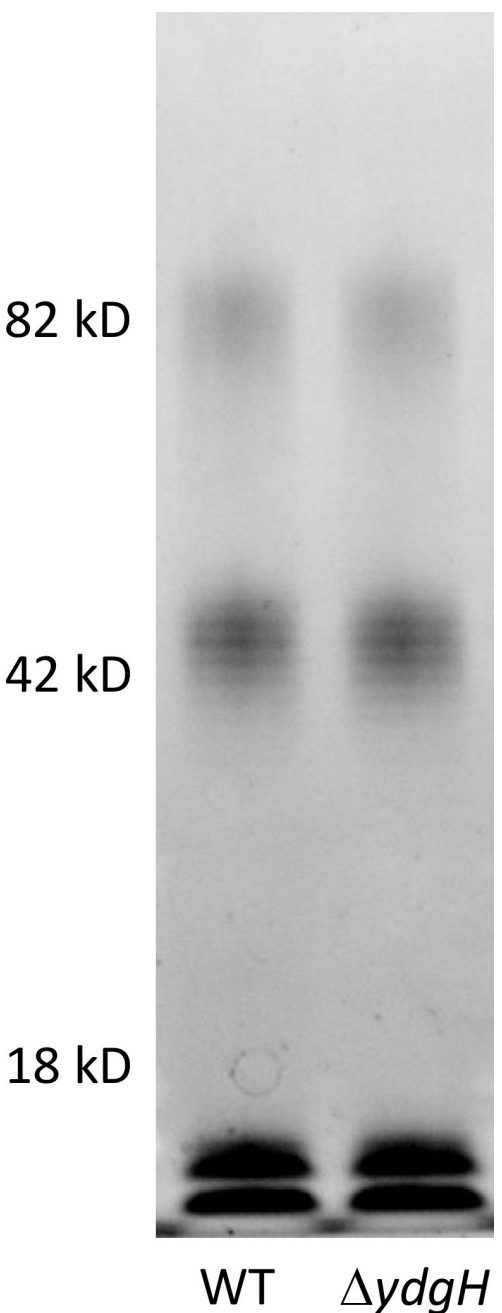

**FIG 7** Lipopolysaccharide fractions from *E. cloacae* ATCC 13047 WT and Δ*ydgH* mutant were extracted, electrophoresed, and stained using a fluorescent LPS stain.

The finding of two independent mutations (predicted to cause frameshift) in *yhdP* is also tantalizing. *yhdP* is a member of a group of AsmA-like paralogs that has been found to participate in phospholipid partitioning between the inner and outer membranes (33). The function of *yhdP* was originally suggested when disruption of it was found to rescue the phospholipid mislocalization phenotype of Mla system mutants (34). Detailed phenotypes of the *ydgH* and *yhdP* double mutants should be investigated, as should double mutants of *ydgH* and other AsmA-like paralogs. Finally, work should be done to characterize the effect of *ydgH* deletion on other established determinants of antibiotic resistance, such as alternative porins and efflux pumps. RNAseq and other hypothesis-generating techniques may be the most efficient path forward to assess for decreases or increases in expression in other candidate mediators. Further structural characterization

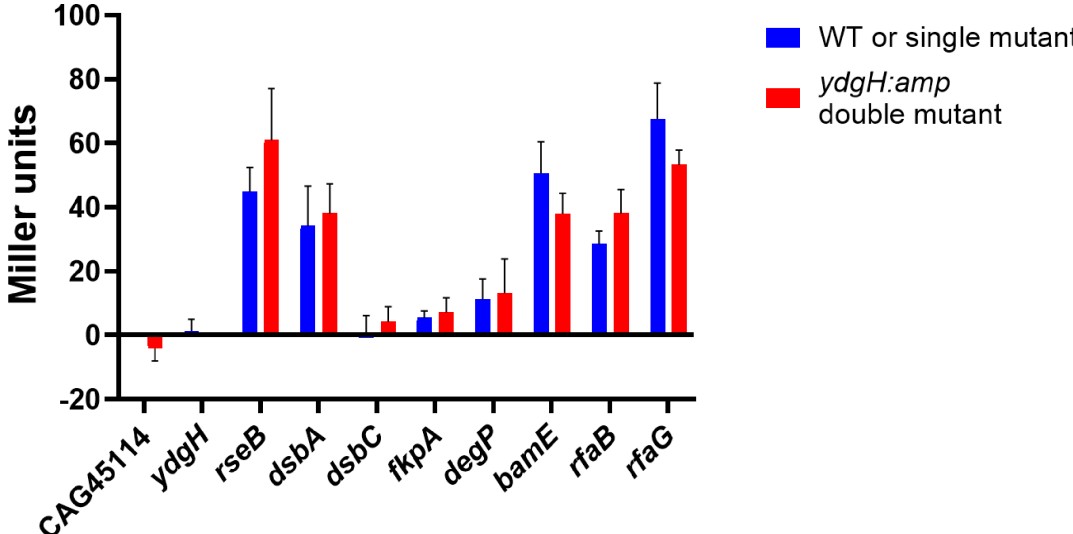

**FIG 8** *ydgH* disruption, neither alone nor in combination with other envelope-related genes, leads to activation of the sigma E response. CAG45114 is an *E. coli* strain with a beta-galactosidase reporter for Sigma E activation. Miller units are indicated for either the single mutant (blue) or the double mutant (with *ydgH* additionally disrupted with an ampicillin cassette, blue).

of YdgH through techniques complementary to crystallization such as small-angle X-ray scattering or cryo-electron microscopy has the potential to identify structural homologs.

In summary, we have characterized YdgH as a periplasmic monomer, with its second and third DUF1471 domains necessary for its function. More work remains to be done as to how YdgH functions in outer membrane homeostasis.

**TABLE 1** Mutations in *E. coli* Δ*ydgH* background with wild-type growth on benzalkonium chloride agar[a,b]

| Mutant | Genomic region | Annotation | Nucleotide #; change | Amino acid change |
|---|---|---|---|---|
| 1 | *proQ* | RNA chaperone, putative ProP translation regular | 1,909,425; C -> A | E123* |
| 2 | *ftsI* | Transpeptidase involved in septal peptidoglycan synthesis | 89,248; A -> C | D450A |
| 3 | *mepM* | Murein DD-endopeptidase, septation protein | 1,935,645; T -> C | D83G |
| 4 | *psrO* | Novel sRNA, function unknown | 3,304,739; IS4 insertion (+12 bp) | –[c] |
| 5 | *fadR* | Fatty acid metabolism regulon transcriptional regulator | 1,230,776; T -> G | L128R |
| 6 | *yhbJ* | Adaptor protein RapZ for GlmZ/GlmY sRNA decase | 3,340,756; C -> T | R95C |
| 7 | *pnp* | Polynucleotide phosphorylase/polyadenylase | 3,302,898; Δ8 bp | – |
| 8 | *yhdP* | DUF3971-AsmA2 domains protein | 3,388,713; Δ1 bp | – |
| 9 | *yhdP* | DUF3971-AsmA2 domains protein | 3,387,876; IS5 insertion (+5 bp) | – |
| 10 | *ykgM* (intergenic) *ykgR* | 50S ribosomal protein L31 type B; alternative L31 utilized during zinc limitation | 308,687; T -> C | – |
| 11 | *ftsA* | ATP-binding cell division protein involved in recruitment of FtsK to Z ring | 101,630; G -> A | E388K |
| 12 | *lpxM* | Myristoyl-acyl carrier protein (ACP)-dependent acyltransferase | 1,933,837; T -> G | Y205S |
| 13 | *arnB* | Uridine 5′-(beta-1-threo-pentapyranosyl-4-ulose diphosphate) aminotransferase | 2,359,716; C -> T | R104C |
| | *envZ* | Sensory histidine kinase in two-component regulatory system with OmpR | 3,528,666; C -> A | D188Y |
| | *malS* | Alpha-amylase | 3,730,976; +2 bp | – |

[a]Nucleotide # is in reference to CP009273.
[b]* denotes a new stop codon.
[c]"-" denotes a frameshift.

## ACKNOWLEDGMENTS

We thank the laboratories of Thomas Bernhardt (Harvard University) for supplying P1 lambda phage and of Carol Gross (University of California, San Francisco) for providing CAG45114.

This work was supported by T32 AI-007061, the Harvard Catalyst Medical Research Investigator Training fellowship, and K08 AI-155830 to J.E.L. and HHMI to M.K.W.

## AUTHOR AFFILIATIONS

[1]Department of Medicine, Division of Infectious Diseases, Massachusetts General Hospital, Harvard Medical School, Boston, Massachusetts, USA
[2]Department of Microbiology, Harvard Medical School, Boston, Massachusetts, USA
[3]Department of Medicine, Division of Infectious Diseases, Brigham and Women's Hospital, Harvard Medical School, Boston, Massachusetts, USA
[4]Howard Hughes Medical Institute, Boston, Massachusetts, USA

## AUTHOR ORCIDs

Jacob E. Lazarus http://orcid.org/0000-0001-9286-3548

## FUNDING

| Funder | Grant(s) | Author(s) |
| --- | --- | --- |
| HHS \| National Institutes of Health (NIH) | K08 AI-155830, T32 AI-007061 | Jacob E. Lazarus |
| Harvard Medical School (HMS) | CMeRiT | Jacob E. Lazarus |
| HHMI | | Matthew K. Waldor |

## AUTHOR CONTRIBUTIONS

Jacob E. Lazarus, Conceptualization, Data curation, Formal analysis, Funding acquisition, Investigation, Methodology, Writing – original draft, Writing – review and editing | Matthew K. Waldor, Conceptualization, Methodology, Resources, Writing – review and editing | David C. Hooper, Methodology, Resources, Supervision, Writing – review and editing

## ADDITIONAL FILES

The following material is available online.

Open Peer Review

**PEER REVIEW HISTORY (review-history.pdf).** An accounting of the reviewer comments and feedback.

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
