## [Reviewer comments · Microbiology Spectrum]

Microbiology Spectrum

Characterization of YdgH: A Mediator of Beta-Lactam Susceptibility in Enterobacterales

Jacob Lazarus, Matthew Waldor, and David Hooper

Corresponding Author(s): Jacob Lazarus, Massachusetts General Hospital

Review Timeline:

Submission Date:	August 1, 2024
Editorial Decision:	October 31, 2024
Revision Received:	November 12, 2024
Accepted:	November 14, 2024

Editor: Krisztina Papp-Wallace

Reviewer(s): Disclosure of reviewer identity is with reference to reviewer comments included in decision letter(s). The following individuals involved in review of your submission have agreed to reveal their identity: Arun S Kharat (Reviewer #2)

Transaction Report:

DOI: <https://doi.org/10.1128/spectrum.01940-24>

Re: Spectrum01940-24 (Characterization of YdgH: A Mediator of Beta-Lactam Susceptibility in Enterobacterales)

Dear Dr. Jacob E Lazarus:

Thank you for the privilege of reviewing your work. Below you will find my comments, instructions from the Spectrum editorial office, and the reviewer comments.

Revision Guidelines

Sincerely,
Krisztina Papp-Wallace
Editor
Microbiology Spectrum

Reviewer #2 (Comments for the Author):

While the manuscript is of general interest and proposed hypothesis is good, I have few comments for authors, stated below:
1. The identification of the *S. marcescens* ydgH gene and its alignment with Enterobacterales homologs seems reasonable. However, the article lacks specificity about which homologs were used and if there was a comprehensive comparison with species beyond Enterobacterales. This would provide more context for evolutionary conservation and functional similarities.

2. The identification of a periplasmic signal sequence using SignalP 6.0 is mentioned without providing additional details on the confidence level or alternative methods used to validate the prediction. It is not clear what experiment authors performed that validates SignalP 6.0 existence?
3. The structural description of YdgH is somewhat speculative. Although AlphaFold and Robetta are state-of-the-art tools, if the predictions are supplemented with actual experimental evidence would be better.
4. The hypothesis regarding YdgH's role in periplasmic homeostasis and outer membrane integrity is presented well, but experimental would make it complete and appreciable. The detergent susceptibility experiments and the sigma E promoter investigation needs intensified.
5. The lack of changes in ompC or ompF expression levels weakens the proposed hypothesis of YdgH's involvement in beta-lactam permeation. This is acknowledged, but no further experiments are described to explore alternative mechanisms.
6. Was the restoration of the cefoxitin phenotype statistically significant, and were the experiments replicated to ensure reproducibility?
7. Could the functionality of YdgH in the cefoxitin phenotype be due to general overexpression of proteins rather than its localization?
8. Given that OmpC and OmpF expression was unaffected, what alternative mechanisms for cefoxitin resistance were considered, such as efflux pumps or alternative porins?

While the manuscript is of general interest and proposed hypothesis is good, I have few comments for authors, stated below:

1. The identification of the *S. marcescens* ydgH gene and its alignment with Enterobacterales homologs seems reasonable. However, the article lacks specificity about which homologs were used and if there was a comprehensive comparison with species beyond Enterobacterales. This would provide more context for evolutionary conservation and functional similarities.
2. The identification of a periplasmic signal sequence using SignalP 6.0 is mentioned without providing additional details on the confidence level or alternative methods used to validate the prediction. It is not clear what experiment authors performed that validates SignalP 6.0 existence?
3. The structural description of YdgH is somewhat speculative. Although AlphaFold and Robetta are state-of-the-art tools, if the predictions are supplemented with actual experimental evidence would be better.
4. The hypothesis regarding YdgH's role in periplasmic homeostasis and outer membrane integrity is presented well, but experimental would make it complete and appreciable. The detergent susceptibility experiments and the sigma E promoter investigation needs intensified.
5. The lack of changes in ompC or ompF expression levels weakens the proposed hypothesis of YdgH's involvement in beta-lactam permeation. This is acknowledged, but no further experiments are described to explore alternative mechanisms.
6. Was the restoration of the ceftioxin phenotype statistically significant, and were the experiments replicated to ensure reproducibility?
7. Could the functionality of YdgH in the ceftioxin phenotype be due to general overexpression of proteins rather than its localization?
8. Given that OmpC and OmpF expression was unaffected, what alternative mechanisms for ceftioxin resistance were considered, such as efflux pumps or alternative porins?

Response to Review:

We thank the reviewer for the careful, constructive, and thorough review, and the editor for her coordination. We appreciate their interest in our work.

Reviewer's comments:

While the manuscript is of general interest and proposed hypothesis is good, I have few comments for authors, stated below:

*1. The identification of the *S. marcescens* ydgH gene and its alignment with Enterobacterales homologs seems reasonable. However, the article lacks specificity about which homologs were used and if there was a comprehensive comparison with species beyond*

We have now provided additional detail in the Methods (lines 74,75) which reads:

*"YdgH homologues were identified using a standard protein BLAST using *S. marcescens* YdgH (WP_033640181). Homologues were not present in non-Enterobacterales...The GenBank entries used were: *S. marcescens* WP_033640181; *Serratia liquefaciens* WP_129940662; *Yersinia enterocolitica* WP_083158798; *Hafnia alvei* WP_072310388; *Pantoea agglomerans* WP_109650442; *Escherichia coli* K12 WP_000769303; *Shigella dysenteriae* WP_128880534; *E. coli* O157:H7 H&_AAG586591; *Citrobacter freundii* WP_038640470; *Salmonella enterica* WP_079974581; *Enterobacter cloacae* WP_088221952; *Raoultella planticola* WP_064793623; *Klebsiella aerogenes* WP_047035392; *Klebsiella pneumoniae* WP_110096565; *Chronobacter sakazakii* WP_110868768."*

2. The identification of a periplasmic signal sequence using SignalP 6.0 is mentioned without providing additional details on the confidence level or alternative methods used to validate the prediction. It is not clear what experiment authors performed that validates SignalP 6.0 existence?

We have provided additional detail in the Results (on line 142) which now reads:

"A high-confidence (98%) periplasmic signal sequence was identified by SignalP 6.0 at residues 1-22."

We have clarified the text (on line 143) to indicate how we approached validation of the SignalP prediction:

"To validate this prediction and to determine if this predicted periplasmic signal sequence mediated periplasmic localization..."

3. The structural description of YdgH is somewhat speculative. Although AlphaFold and Robetta are state-of-the-art tools, if the predictions are supplemented with actual experimental evidence would be better.

We have updated the Results to acknowledge the speculative nature of structure prediction tools more directly like AlphaFold and Robetta. Starting on line 162, the text now reads:

“To begin, we modeled the overall polypeptide in AlphaFold, which predicted that there was a relatively large region without predicted structure between DUF1471-B and DUF1471-C (13). As an initial hypothesis of the possible conformations the YdgH DUF1471 domains might make around this unstructured region, we explored possible structures in Robetta (14).”

We have also updated the Discussion to emphasize this as well. The relevant section (starting on line 257) now reads:

“In size exclusion chromatography and cross-linking experiments, we show that YdgH is monomeric, and our early analysis here leads us to speculate that it may form a compact, rather than a more elongated tertiary structure.”

We have also proposed future directions to more precisely define this (starting on line 299):

“Further structural characterization of YdgH through techniques complementary to crystallization such as small angle X-ray scattering or cryo-electron microscopy has the potential to identify structural homologues.”

4. The hypothesis regarding YdgH's role in periplasmic homeostasis and outer membrane integrity is presented well, but experimental would make it complete and appreciable. The detergent susceptibility experiments and the sigma E promoter investigation needs intensified.

5. The lack of changes in ompC or ompF expression levels weakens the proposed hypothesis of YdgH's involvement in beta-lactam permeation. This is acknowledged, but no further experiments are described to explore alternative mechanisms.

We agree with the reviewer about the need for future experiments to explore alternative mechanisms. In the Discussion section (starting on line 279), we propose investigation of peptidoglycan structure and the muropeptide pool in $\Delta ydgH$, and exploration of genetic interaction between *ydgH* and AsmA-like paralogs. To address the question about outer membrane integrity, we have suggested future experiments to address this more directly (on line 275):

“Additional experiments that could be performed to assess for mild defects in outer membrane integrity, either in single or in compound mutants include fluorescent dye penetration and periplasmic enzyme leakage assays.”

6. Was the restoration of the cefoxitin phenotype statistically significant, and were the experiments replicated to ensure reproducibility?

All depicted experiments were replicated to ensure reproducibility. The restoration of the cefoxitin phenotype in Figure 5 was statistically significant ($p < 0.001$ using unpaired t tests) for the full-length YdgH construct and the 101C construct.

7. Could the functionality of YdgH in the cefoxitin phenotype be due to general overexpression of proteins rather than its localization?

8. Given that OmpC and OmpF expression was unaffected, what alternative mechanisms for cefoxitin resistance were considered, such as efflux pumps or alternative porins?

We thank the reviewer for these comments. We have altered the manuscript in the Discussion (on line 295) to read:

“Finally, work should be done to characterize the effect of ydgH deletion on other established determinants of antibiotic resistance, such as alternative porins and efflux pumps. RNAseq and other hypothesis-generating techniques may be the most efficient path forward to assess for decreases or increases in expression in other candidate mediators.”

Re: Spectrum01940-24R1 (Characterization of YdgH: A Mediator of Beta-Lactam Susceptibility in Enterobacterales)

Dear Dr. Jacob E Lazarus:

Your manuscript has been accepted, and I am forwarding it to the ASM production staff for publication. Your paper will first be checked to make sure all elements meet the technical requirements. ASM staff will contact you if anything needs to be revised before copyediting and production can begin. Otherwise, you will be notified when your proofs are ready to be viewed.

Sincerely,
Krisztina Papp-Wallace
Editor
Microbiology Spectrum